# A country-wide survey of knowledge and practice regarding prevention, snake identification and pre-hospital management of children with snakebites amongst Public Health Midwives in Sri Lanka

Kavinda Dayasiri[1]*, Tharuka Perera[1], Gihan Gunarathna[1], Achila Ranasinghe[1], Indika Gawarammana[2], Shaluka Jayamanne[3]

1 Department of Paediatrics, Faculty of Medicine, University of Kelaniya, Ragama, Sri Lanka,
2 Depaertment of Medicine, Faculty of Medicine, University of Peradeniya, Peradeniya, Sri Lanka,
3 Department of Paediatrics, Faculty of Medicine, University of Kelaniya, Ragama, Sri Lanka

* kavindadayasiri@gmail.com

## Abstract

### Background

Snake envenoming remains a major public health problem in Sri Lanka, disproportionately affecting rural children. Timely and appropriate pre-hospital management is crucial to reducing morbidity and mortality. Public Health Midwives (PHMs), as front-line community health workers, are strategically positioned to contribute to snakebite prevention and early management. However, their knowledge and confidence regarding paediatric snakebite care have not been systematically evaluated.

### Methodology/Principal findings

A cross-sectional, country-wide survey was conducted among PHMs across all 25 districts of Sri Lanka using a stratified random sampling approach. Data were collected through a validated, self-administered online questionnaire designed to assess knowledge and practices related to paediatric snakebite prevention, snake identification, and pre-hospital management. A total of 1,706 PHMs participated. Although most respondents correctly identified essential first-aid measures such as reassuring the patient (95.9%) and prompt hospital transfer (95.8%), a substantial proportion endorsed harmful practices, including sucking the venom (71.6%), applying herbs (73.2%), or cutting the bite site (64.4%). Only 48.7% recognized immobilisation as an appropriate first aid. Snake identification accuracy was high for the cobra (94%) but low for other medically significant species such as kraits and vipers. Awareness of national guidelines (14.7%) and emergency contact services (11.9%) was also poor. PHMs with prior training, awareness of national information services, or previous

**Data availability statement:** All data are in the manuscript and/or supporting information files.

**Funding:** This study was supported by the University of Kelaniya Research Council Grant No RC/2025/PPCR02 awarded to KD. TP received salary support from the University of Kelaniya Research Council Grant (RC/2025/PPCR02). The funders had no role in study design, data collection and analysis, decision to publish, or preparation of the manuscript.

**Competing interests:** The authors have declared that no competing interests exist.

experience providing first aid demonstrated significantly higher knowledge scores (p < 0.001).

## Conclusions/Significance

Despite their key role in community health, PHMs in Sri Lanka exhibit major knowledge gaps in evidence-based first aid, snake identification, and awareness of national support systems for snakebite management. These deficits, coupled with widespread misconceptions and low confidence, highlight the need for structured training, curriculum integration, and ongoing professional development. Strengthening PHMs' capacity in snakebite prevention and early management can significantly enhance community preparedness and reduce the burden of paediatric snakebite envenoming in Sri Lanka.

## Author summary

Snakebites cause substantial illness and death in Sri Lanka, particularly among children in rural areas. Public Health Midwives (PHMs) are the main community-based health workers in the country and play a vital role in health education and emergency response. This nationwide study, conducted among 1,706 PHMs, explored their knowledge and practices related to snakebite prevention, identification, and pre-hospital management in children. We found that while most PHMs understood the importance of calming the victim and taking the child promptly to a hospital, many continued to endorse traditional but harmful first-aid practices such as applying herbs and tourniquets, making cuts, or sucking out venom. Very few were aware of national snakebite guidelines or emergency information services. Those who had received any form of prior training were more confident and demonstrated better knowledge. The study highlights a critical need for national training programs, inclusion of snakebite modules in PHM education, and integration of snakebite awareness into community health promotion. Empowering PHMs with accurate knowledge and confidence can improve early management, reduce complications, and save lives of children affected by snakebites in Sri Lanka.

## Introduction

Snake envenoming is a neglected tropical disease that continues to pose a major public health challenge in Sri Lanka, particularly among children living in rural and agricultural communities [1]. The country records thousands of snakebite cases annually, with children often representing a disproportionately affected group due to their limited awareness, outdoor activity patterns, and dependence on adult supervision for timely medical care [2]. The morbidity and mortality associated with paediatric snakebites are largely preventable through prompt recognition, appropriate

first-aid, and early hospital transfer [3,4]. However, the persistence of harmful traditional practices, delays in seeking care, and insufficient knowledge among first responders continue to compromise outcomes [4,5,6].

Public Health Midwives (PHMs) constitute the backbone of Sri Lanka's primary healthcare system and are the most widely distributed category of community health workers [7]. Their close engagement with families through maternal and child health programmes uniquely positions them to educate communities about snakebite prevention, identify risk factors, and facilitate appropriate pre-hospital management. Despite this, formal training in snakebite management has not traditionally been included in community health worker curricula, and data regarding their knowledge, confidence, and practices remain scarce [8].

Previous studies in Sri Lanka have predominantly focused on hospital-based management of snakebites and clinical outcomes, with little attention to community-level preparedness or preventive education [9]. Evidence from other low- and middle-income countries suggests that health workers' inadequate knowledge of snakebite first-aid and reliance on traditional remedies contribute to delayed treatment and adverse outcomes [10]. In the Sri Lankan context, where healthcare access and health-seeking behaviour are strongly influenced by community-level health workers, assessing PHMs' knowledge and practices represents an essential step toward strengthening pre-hospital care and prevention of paediatric snakebites.

This study, therefore, aimed to assess the knowledge and practices of Public Health Midwives across Sri Lanka regarding the prevention, identification, and pre-hospital management of paediatric snakebites. By identifying gaps and determinants of knowledge, the findings are expected to inform national capacity-building initiatives, guide curriculum development, and contribute to the integration of community-based snakebite prevention strategies within the existing primary healthcare framework.

## Materials and methods

### Ethics statement

Ethical approval for the study was obtained from the Ethics Review Committee, Post-graduate Institute of Medicine, University of Colombo (Reference: ERC/PGIM/2024/080), and administrative clearance was granted by the Ministry of Health, Sri Lanka (Reference: ETR/G/AC/16/2025). The objectives, procedures, and voluntary nature of participation were clearly explained to each participant at the beginning of the online form, and formal written informed consent was obtained prior to participation. No personally identifiable information was collected, and all data were handled with strict confidentiality. The study was supported by a University of Kelaniya Research Council Grant (RC/2025/PPCR02) awarded to Professor Kavinda Dayasiri. The research was conducted in collaboration with the Family Health Bureau and the Education, Training and Research Unit of the Ministry of Health, Sri Lanka.

### Study design

This observational cross-sectional study assessed the knowledge and practices of Public Health Midwives in Sri Lanka regarding the prevention, snake identification, and pre-hospital management of paediatric snakebites. The cross-sectional approach was selected to provide a comprehensive snapshot of current awareness and field practices across all 25 administrative districts of the country. PHMs play a central role in Sri Lanka's primary healthcare system, particularly in community-level health promotion, maternal and child health, and emergency response; thus, their understanding and preparedness are critical to community snakebite management and prevention efforts.

### Participant recruitment and selection

The study population comprised PHMs currently employed under the Ministry of Health, Sri Lanka, who were engaged in direct community work. Inclusion criteria encompassed all PHMs actively involved in field-level maternal and child health

services, community education, and pre-hospital care. PHMs serving exclusively in administrative or hospital-based roles without direct public interaction were excluded. Participation was entirely voluntary, and only those providing informed consent were included in the analysis.

To ensure broad national representativeness, a stratified random sampling method was used. Each administrative district was treated as a stratum, and PHMs were randomly selected from official lists maintained by Medical Officer of Health (MOH) offices. The number of participants from each district was determined proportionate to its total PHM cadre, thereby maintaining a balanced sample distribution. The minimum sample size was calculated using the formula, assuming a 95% confidence level (Z = 1.96), 5% precision (d = 0.05), and an expected proportion of adequate knowledge of 50% to maximize the sample. Accounting for a 20% non-response rate, a minimum of 460 participants was required, providing over 80% power to detect differences in knowledge levels among Public Health Midwives. The study however, achieved a high national response rate, with a considerably larger number of PHMs (n = 1706) participating across all districts, enhancing the precision and external validity of the findings.

## Data collection procedures

A multi-structured, self-administered questionnaire was designed specifically for this study to capture PHMs' knowledge and practices related to paediatric snakebites. The instrument consisted of four major sections. The first section collected demographic and professional information, including age, district, years of service, and previous training in snakebite management. The second section assessed knowledge and attitudes toward first-aid and pre-hospital care, including awareness of recommended measures such as immobilisation and reassurance, and recognition of potentially harmful traditional practices such as cutting, sucking, or applying tourniquets and herbal substances to the bite site. The third section assessed snake identification ability using high-resolution photographs of common Sri Lankan snake species, including both venomous and non-venomous types such as the cobra, Russell's viper, hump-nosed viper, green pit viper, kraits, python, rat snake, and sea snake. Participants were asked to identify the species and indicate whether they believed it to be venomous or non-venomous. The final section evaluated knowledge on prevention strategies, recognition of envenomation signs, potential complications, and general awareness of treatment principles and national guidelines for snakebite management.

The questionnaire was developed in English and translated into Sinhala and Tamil, ensuring accessibility for participants across all regions. Forward and backward translation procedures were used to maintain conceptual equivalence. The preliminary draft was reviewed by an expert panel comprising a consultant paediatrician, a medical toxicologist, and a consultant psychiatrist to ensure content validity, clarity, and cultural appropriateness. The tool was subsequently pilot-tested with ten PHMs (five Sinhala-speaking and five Tamil-speaking) who provided feedback on question clarity and structure. Modifications were incorporated based on this feedback to improve comprehension and relevance

Data collection was conducted over a six-month period using a Google Form–based online questionnaire distributed via MOH offices in each district. The link to the form was shared with PHMs through official text messages or emails. For those with limited access to smartphones or the internet, data collection was facilitated using digital devices available at MOH offices. The data collection process was entirely contactless; investigators did not directly interact with participants, thereby minimising interviewer bias. The questionnaire was anonymous, and responses were automatically uploaded to a secure database accessible only to the research team.

## Data analysis

Data were exported from the Google database into Microsoft Excel and analyzed using IBM SPSS Statistics, Version 26. Initial data cleaning involved checking for incomplete entries and duplicate responses. Descriptive statistics were used to summarize demographic and professional characteristics, as well as responses related to knowledge, attitudes, and practices. Each correct response was assigned a score of one point, and the total knowledge score was calculated for

each participant. Based on the distribution of scores, participants were categorized into two groups—those with adequate knowledge (≥75th percentile) and inadequate knowledge (<75th percentile).

Bivariate analyses were performed using the Chi-square test ($\chi^2$) to explore associations between the adequacy of knowledge and independent variables such as training history, previous exposure to snakebite cases, awareness of national guidelines, and self-reported confidence in managing snakebites. Statistical significance was set at a p-value of less than 0.05.

Multiple strategies were employed to maintain data quality and reduce bias. Automated online data entry minimized transcription errors. The use of standardized instructions and a validated instrument improved reliability, while stratified random sampling reduced selection bias and enhanced the representativeness of the sample. Anonymous participation was encouraged to minimise social desirability bias, ensuring that responses reflected genuine knowledge and beliefs.

## Results

### Participant characteristics

The survey included 1706 Public Health Midwives across all 25 administrative districts of the country. Mean age of PHMs was 42 years (SD- 4.18, range: 22–62 years). All PHMs were female and engaged in field-level maternal and child health services. Only a small proportion reported prior involvement in snakebite management, with 60 (3.5%) having given first aid to a child with a snakebite and 77 (4.5%) having received formal training for this. Awareness of national resources was also limited: 203 (11.9%) had heard of a national telephone number for snakebite emergencies, 130 (7.6%) knew about a national information service for identifying the biting snake, and 250 (14.7%) were aware of a national guideline for snakebite treatment. Despite this, the vast majority (1,631; 95.6%) recognized the usefulness of first aid training for children with snakebites, although only 350 (20.5%) felt confident in providing such first aid themselves.

### First-aid management

Most PHMs correctly recognised the importance of reassuring and calming the victim (95.9%) and transporting the child promptly to a hospital with antivenom facilities (95.8%). However, widespread endorsement of traditional and potentially harmful practices was observed. A majority indicated that they would suck out the venom (71.6%), apply herbs (73.2%), or make an incision at the bite site (64.4%). Approximately, 40% endorsed applying a tourniquet as an appropriate first-aid technique, while 59.9% stated they would not recommend any first aid at all. Overall, fewer than half of the respondents demonstrated adequate first-aid knowledge according to the scoring criteria (Table 1).

### Associations of first-aid knowledge

Bivariate analysis showed that PHMs who had previously received first-aid training, were aware of national information services or treatment guidelines, or had higher confidence in managing snakebites were significantly more likely to possess adequate knowledge (p < 0.001). Those who considered first-aid training useful also scored higher, indicating a positive association between motivation for training and knowledge adequacy (Table 2).

### Snake identification and venomous species recognition

The ability to identify snakes from high-resolution coloured photographs varied substantially across snakes. Recognition was highest for the Indian cobra (94%) and green pit viper (74.9%), but very low for other medically important snakes such as the Indian krait (20.1%), Ceylon krait (28.2%), and Russell's viper (33.7%). Although over 80% correctly identified the Indian cobra, Russell's viper, and hump-nosed viper as venomous, fewer than one-third recognised the venomous nature of sea snakes (Fig 1).

**Table 1. Knowledge of first-aid management practices for paediatric snakebites among Public Health Midwives (n = 1,706). (PHM – Public Health Midwife).**

| Action | Correct response | PHMs answering correctly - n (%) (n = 1706) |
|---|---|---|
| Cutting the bite site | Not recommended | 1098(64.4%) |
| Sucking venom from wound | Not recommended | 1221(71.6%) |
| Applying a tourniquet | Not recommended | 692(40.6%) |
| Applying ice | Not recommended | 994(58.3%) |
| Squeezing blood out of the bite wound | Not recommended | 828(48.5%) |
| Washing gently with water | Recommended | 1464(85.8%) |
| Removing tight items (jewellery) | Recommended | 1314(77.0%) |
| Immobilisation of limb | Recommended | 831(48.7%) |
| Massaging the bitten site | Not recommended | 1305(76.5%) |
| Applying herbs | Not recommended | 1249(73.2%) |
| Advising not to panic | Recommended | 498(29.2%) |
| Giving a toxin lowering drink | Not recommended | 928(54.4%) |
| Applying surgical spirit | Not recommended | 930(54.5%) |
| Reassuring and calming child | Recommended | 1636(95.9%) |
| Immediate transport to hospital | Recommended | 1634(95.8%) |
| Not giving any first-aid | Not recommended | 1022(59.9%) |

**Table 2. Association between selected characteristics and adequacy of knowledge on snakebite first aid among Public Health Midwives (Adequate knowledge: sub-total ≥11; Inadequate knowledge: sub-total ≤10).**

| | Adequate knowledge (N = 1343) | Inadequate knowledge (N = 363) | $X^2$ | P value |
|---|---|---|---|---|
| Trained to give first aid | 76(5.6%) | 1(0.2%) | 19.21 | <0.001 |
| Involved in giving first aid to a child with snakebite | 54(4.0%) | 6(1.6%) | 4.72 | 0.30 |
| Ever heard of a national information service | 121(9.0%) | 9(2.4%) | 17.12 | <0.001 |
| Ever heard of an emergency contact number | 181(13.4%) | 22(6.0%) | 14.99 | <0.001 |
| Ever heard of a national guideline on snakebite management | 219(16.3%) | 31(8.5%) | 13.78 | <0,001 |
| Think first-aid training is useful | 1295(96.4%) | 336(92.5%) | 10.31 | 0.001 |
| Confident in managing snakebites | 1096(81.6%) | 260(71.6%) | 14.46 | <0.001 |

Overall, the average accuracy of snake identification across all species was below 50%. PHMs with adequate first-aid knowledge were significantly more likely to correctly identify snakes and to be aware of their venomous status (p < 0.001). Table 3 summarizes factors associated with the ability of Public Health Midwives to correctly identify snakes. Those with adequate knowledge of snakebite first-aid were significantly more likely to have received first-aid training, to be aware of national information services and guidelines, and to have prior experience providing first aid to a child with snakebite. Confidence in managing snakebites was also significantly higher among those demonstrating adequate identification skills.

### Knowledge of snakebite signs, complications, and treatment

Most PHMs demonstrated satisfactory understanding of the general features of envenomation. The majority recognized that envenomation can be confirmed by a blood test (81.7%), that effective treatment is available (90.5%), and that snakebite risk varies by season (63.1%). However, major misconceptions persisted. Nearly half believed that

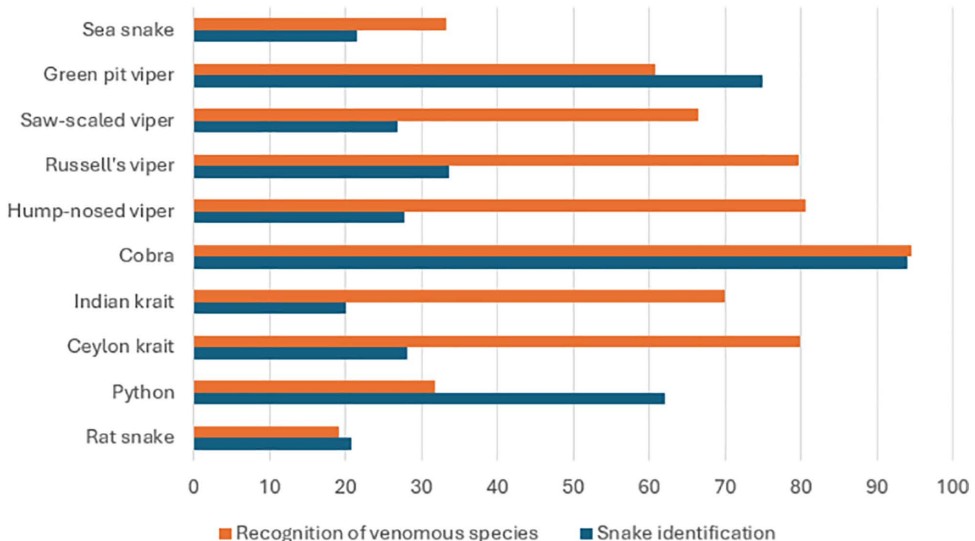

**Fig 1. Variation in the percentage of correct responses for snake identification and venomous species recognition.**

Table 3. Associations of snake identification by Public Health Midwives (Adequate knowledge: sub-total ≥10; Inadequate knowledge: sub-total ≤9).

| | Adequate knowledge (N = 1061) | Inadequate knowledge on (N = 645) | X² | P value |
|---|---|---|---|---|
| Trained to give first aid | 64(6.31%) | 13(2.01%) | 15.01 | <0.001 |
| Involved in giving first aid to a child with snakebite | 46(4.33%) | 14(2.1%) | 5.54 | 0.019 |
| Ever heard of a national information service | 102(9.6%) | 28(4.3%) | 15.84 | <0.001 |
| Ever heard of an emergency contact number | 136(12.8%) | 67(10.3%) | 2.26 | 0.133 |
| Ever heard of a national guideline on snakebite management | 192(18.0%) | 58(8.9%) | 26.58 | <0.001 |
| Think first-aid training is useful | 1022(96.3%) | 609(94.4%) | 3.49 | 0.063 |
| Confident in managing snakebites | 839(79.0%) | 517(80.1%) | 41.28 | <0.001 |

antivenom should be given to all patients bitten by snakes (49.9%), and only half (50.8%) were aware that allergic reactions to antivenom are common. Recognition of snake-specific complications was poor—only 34.2% associated acute kidney injury with the hump-nosed viper, and less than 10% identified it as a complication of krait bites. Similarly, only 38.2% associated muscle paralysis with cobra envenomation, and 29.7% recognised bleeding as a feature of Russell's viper bites (Table 4).

## Overall, knowledge scores and associated factors

Using the composite knowledge score, 58% of PHMs were categorized as having adequate knowledge. Adequate knowledge was significantly associated with prior first-aid training (p < 0.001), previous experience managing a child with snakebite (p < 0.001), awareness of national guidelines and emergency contact services (p < 0.001), and higher self-reported confidence in management (p < 0.001). Those who perceived first-aid training as useful also demonstrated significantly better knowledge levels. These findings emphasize that experiential learning and structured training strongly predict knowledge adequacy among PHMs (Table 5).

**Table 4. Knowledge regarding signs, complications, and treatment of paediatric snakebites among Public Health Midwives.**

| Knowledge regarding snake bites in children | Correct response | Number (%) of correct responses (n = 1706) |
|---|---|---|
| All bites by venomous snakes lead to envenomation | False | 217(12.7%) |
| Fang marks differentiate venomous and nonvenomous snakes | False | 406(23.8% |
| The risk of snake bites is more in certain months of the year | True | 1077(63.1%) |
| Envenomation is confirmed by a blood test | True | 1394(81.7%) |
| Knowledge regarding treatment of snakebites | | |
| There is effective treatment available for venomous snakebites | True | 1544(90.5%) |
| Allergies to antivenom is common in Sri Lanka | True | 867(50.8%) |
| Antivenom are the only specific antidotes in the management of snake bites | True | 839(49.2%) |
| Antivenoms made anywhere in the world is appropriate for all countries | False | 554(32.5%) |
| Antivenoms should be given to all patients bitten by snakes | False | 732(49.9%) |
| Antivenom therapy is effective for envenomation | True | 957(57.1%) |
| Treatment of bite related pain is similar to any other type of pain | False | 797(46.1%) |
| Signs and symptoms of envenomation | | |
| Prominent fang marks | False | 224(13.1%) |
| Multiple bite marks | False | 275(16.1%) |
| Vomiting | True | 1265(74.2%) |
| Sweating | True | 1188(69.8%) |
| Low blood pressure | True | 1282(75.2%) |
| Complications following a snakebite | | |
| Bleeding - Russell's viper | True | 506(29.7%) |
| Muscle paralysis – Cobra | True | 651(38.2%) |
| Local ulceration - Cobra | True | 390(22.9%) |
| Acute kidney injury - Ceylon krait | False | 163(9.6%) |
| Acute kidney injury – Hump-nosed viper | True | 584(34.2%) |

## Perceived challenges in managing paediatric snakebites

PHMs identified multiple systemic and sociocultural challenges limiting effective pre-hospital management. The most frequently cited barriers included lack of exposure to formal training (96.6%), transport difficulties (93.1%), absence of nearby healthcare facilities (87.5%), and strong traditional beliefs among parents (89.3%). More than four-fifths reported limited clinical exposure during midwifery training (81.7%) and lack of financial or institutional support for community-level education programs. Over half (52.1%) perceived that their efforts in snakebite prevention were inadequately appreciated by health authorities (Table 6).

## Discussion

This nationwide survey provides the first comprehensive assessment of Public Health Midwives' (PHMs) knowledge and practices concerning paediatric snakebites in Sri Lanka. Findings reveal that while most PHMs possessed basic

**Table 5. Associations of overall knowledge regarding paediatric snakebites among Public Health Midwives (Adequate knowledge: total ≥29; Inadequate knowledge: sub-total ≤28).**

|  | Adequate knowledge (N=989) | Inadequate knowledge on (N=717) | X² | P value |
|---|---|---|---|---|
| Trained to give first-aid | 71(7.1%) | 6(0.8%) | 38.79 | <0.001 |
| Involved in giving first aid to a child with snakebite | 52(5.2%) | 8(1.1%) | 21.01 | <0.001 |
| Ever heard of a national information service | 101(10.2%) | 29(4.04%) | 22.46 | <0.001 |
| Ever heard of an emergency contact number | 142(14.3%) | 61(8.50%) | 13.57 | <0.001 |
| Ever heard of a national guideline on snakebite management | 187(18.9%) | 63(8.78%) | 34.04 | <0.001 |
| Think first aid training is useful | 968(97.8%) | 673(93.8%) | 8.91 | 0.003 |
| Confident of managing snakebites | 877(88.6%) | 479(68.8%) | 17.33 | <0.001 |

**Table 6. Reported challenges faced by Public Health Midwives in the prevention and pre-hospital management of paediatric snakebites.**

|  | n (%) |
|---|---|
| Transport difficulties | 1589(93.1%) |
| No nearby health care facilities | 1493(87.5%) |
| Traditional beliefs and rituals among parents | 1524(89.3%) |
| Lack of exposure to effective training programs | 1614(96.6%) |
| Lack of clinical exposure during the midwifery training program | 1394(81.7%) |
| Lack of financial resources | 1372(80.4%) |
| Lack of support from the serving community | 1010(59.2%) |
| Lack of appreciation of the service by the health authority | 888(52.1%) |

awareness of the importance of early hospital transfer and the need to reassure and immobilize the patient, a considerable proportion endorsed harmful traditional practices such as sucking venom, applying herbs, or cutting the bite site. These findings underscore a critical gap between correct knowledge and actual beliefs that could influence field practices, particularly in rural areas where PHMs serve as primary sources of emergency advice.

The limited confidence reported by many PHMs in administering first aid for snakebites highlights a major capacity deficit in the pre-hospital management chain. Although more than 95% of respondents agreed that training would be useful, fewer than 5% had ever received formal instruction in snakebite first aid. This aligns with previous reports from South Asia and sub-Saharan Africa, where inadequate training among frontline health workers has been identified as a major barrier to effective snakebite management [11,12]. The absence of structured training within Sri Lanka's public health system likely contributes to reliance on anecdotal practices and inconsistent community messaging.

The results further demonstrate poor snake identification skills, particularly for medically important species such as kraits and vipers. While cobras and green pit vipers were recognised by most respondents, photographic identification accuracy for other species remained below 40%. This is concerning given that accurate recognition of venomous snakes aids in risk assessment, targeted education, and timely referral decisions [13]. Awareness of national information services, emergency contact numbers, and guidelines on snakebite management was also notably low, indicating a lack of integration between existing policy frameworks and ground-level public health services.

Associations of adequate knowledge—such as prior training, previous experience managing snakebites, awareness of national guidelines, and self-reported confidence—suggest that knowledge gaps are modifiable through structured educational interventions. PHMs who had previously attended training or had direct exposure to snakebite management scored

significantly higher across all domains, supporting the need for regular in-service training programs and inclusion of snakebite modules in midwifery curricula. Beyond conventional training programmes, addressing the gaps identified in this study requires innovative, system-level, and technology-supported approaches. Given PHMs' widespread use of mobile phones, integrating snakebite decision-support tools into existing digital health platforms could provide real-time guidance on first aid, danger signs, and referral pathways. Mobile-based visual aids and short video modules in local languages could help PHMs deliver consistent, culturally sensitive education that counters harmful traditional practices without dismissing community beliefs [14]. In parallel, collaboration with local community leaders, school systems, and agricultural networks could support community-wide risk communication campaigns that align biomedical advice with local contexts.

Importantly, these capacity-building efforts should be designed to directly address the structural and sociocultural barriers identified by PHMs. In settings where transport delays and limited facility access hinder timely care, decision-support resources that function at community level can help PHMs prioritise early immobilisation, rapid referral, and clear communication with families. Integration with national poison information services or broader digital health platforms may further enable real-time consultation, secure information sharing, and rapid referral guidance, particularly in remote areas where transport delays are common [15]. At the same time, educational strategies must acknowledge the strong influence of traditional beliefs reported by participants [16,17,18]. Combining structured training with culturally sensitive communication approaches, supported by locally relevant visual materials, may help PHMs counter harmful practices without alienating communities [12]. Embedding these approaches within existing maternal and child health outreach activities would allow snakebite education to become a routine part of community engagement rather than a standalone intervention.

Beyond individual-level factors, the study highlights several systemic challenges. Over 90% of PHMs cited logistical barriers such as transport difficulties, lack of nearby healthcare facilities, and strong community adherence to traditional beliefs as major obstacles to effective management. These findings mirror broader rural healthcare inequities and emphasize the necessity of combining training with community awareness and system-level improvements [14]. PHMs' close relationship with families and village health committees provides a unique opportunity to deliver culturally sensitive health education and counter harmful myths surrounding snakebites [19].

The widespread belief in traditional remedies, despite correct theoretical understanding of modern first aid, suggests cognitive dissonance between formal training and cultural norms. Behavioural change communication targeting both PHMs and the communities they serve is therefore critical [20]. Empowering PHMs with contextually tailored educational materials and visual aids could enhance their ability to translate knowledge into consistent, evidence-based community practices [21]. Behavioural and psychological factors—such as risk perception, social norms, cultural beliefs, and confidence in challenging traditional advice—likely influence how PHMs translate knowledge into action. Training programmes should therefore incorporate behaviour change communication techniques, including scenario-based learning, role-play, and strategies for addressing misconceptions in culturally sensitive ways. Strengthening PHMs' skills in persuasive health communication and community engagement may help bridge the gap between awareness and practice, ensuring that evidence-based first-aid measures are consistently promoted and adopted at household level.

Comparatively, similar studies among community health workers and rural communities in India and Bangladesh have reported congruent patterns of limited formal training, poor snake identification ability, and reliance on traditional first aid methods [22,23,24]. These regional consistencies reinforce the global recognition of snakebite envenoming as a neglected tropical disease requiring community-based interventions. The present study provides a national evidence base to support Sri Lanka's inclusion of snakebite awareness within child health programs and the broader One Health approach to venomous animal bites.

This study has several limitations. First, the cross-sectional design provides a snapshot of knowledge and practices at a single point in time and does not allow for causal inferences regarding factors associated with knowledge levels. Second, data were collected using a self-administered online questionnaire, and responses may therefore be subject to recall bias and social desirability bias, despite efforts to ensure anonymity and minimize response pressure. Participants may

have reported what they perceived to be desirable or expected answers rather than their true beliefs or intended actions. Furthermore, the findings reflect self-reported knowledge and stated practices rather than direct observation of real-world behaviour. The study did not assess actual first-aid practices performed in field settings, and therefore discrepancies may exist between reported knowledge and real-life responses during snakebite emergencies. Although stratified sampling ensured representation from all districts, participation required access to digital communication platforms; PHMs with limited internet or smartphone access may have been underrepresented. Despite these limitations, the large sample size and nationwide coverage enhance the reliability and generalizability of the findings.

## Conclusions

This country-wide assessment demonstrates that Public Health Midwives in Sri Lanka possess moderate theoretical knowledge but limited practical competence and confidence regarding the pre-hospital management of paediatric snakebites. Persistent endorsement of harmful practices and inadequate snake identification highlight urgent training needs. Structured national capacity-building initiatives, integration of snakebite education into PHM curricula, and strengthened linkages with the national poison information services are essential to improving community-level responses. Empowering PHMs through training, educational materials, and clear referral pathways could transform them into effective advocates for snakebite prevention and early management, thereby reducing morbidity and mortality among Sri Lankan children.

Looking ahead, strengthening community-level snakebite management could also benefit from integration with emerging digital health systems that enable secure information sharing, rapid access to national guidelines, and improved coordination between frontline workers and referral centres. Aligning PHM training with such evolving digital public health infrastructure may further enhance timely decision-making and continuity of care, particularly in geographically remote settings.

## Supporting information

**S1 File. Anonymous database.**
(XLSX)

## Acknowledgments

The authors would like to thank the participants for their time and willingness to share their experiences and insights.

## Author contributions

**Conceptualization:** Kavinda Dayasiri.

**Data curation:** Kavinda Dayasiri.

**Formal analysis:** Kavinda Dayasiri, Gihan Gunarathna.

**Funding acquisition:** Kavinda Dayasiri.

**Investigation:** Kavinda Dayasiri, Tharuka Perera, Gihan Gunarathna, Achila Ranasinghe.

**Methodology:** Kavinda Dayasiri.

**Project administration:** Kavinda Dayasiri.

**Supervision:** Indika Gawarammana, Shaluka Jayamanne.

**Validation:** Kavinda Dayasiri.

**Writing – original draft:** Kavinda Dayasiri.

**Writing – review & editing:** Kavinda Dayasiri.

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
