## [Decision Letter · Decision Letter 0]

28 Jan 2026

A country-wide survey of knowledge and practice regarding prevention, snake identification and pre-hospital management of children with snakebites amongst Public Health Midwives in Sri Lanka

Dear Dr. Dayasiri,

Thank you for submitting your manuscript to PLOS Neglected Tropical Diseases. After careful consideration, we feel that it has merit but does not fully meet PLOS Neglected Tropical Diseases's publication criteria as it currently stands. Therefore, we invite you to submit a revised version of the manuscript that addresses the points raised during the review process.

Please submit your revised manuscript within by Mar 29 2026 11:59PM. If you will need more time than this to complete your revisions, please reply to this message or contact the journal office at plosntds@plos.org. Please include the following items when submitting your revised manuscript:

We look forward to receiving your revised manuscript.

Kind regards,

Wuelton Monteiro, Ph.D.

Section Editor

Wuelton Monteiro

Section Editor

Shaden Kamhawi

co-Editor-in-Chief

Paul Brindley

co-Editor-in-Chief

**Journal Requirements:**

At this stage, the following Authors/Authors require contributions: Kavinda Chandimal Dayasiri. Please ensure that the full contributions of each author are acknowledged in the "Add/Edit/Remove Authors" section of our submission form.

2) Some material included in your submission may be copyrighted. According to PLOSu2019s copyright policy, authors who use figures or other material (e.g., graphics, clipart, maps) from another author or copyright holder must demonstrate or obtain permission to publish this material under the Creative Commons Attribution 4.0 International (CC BY 4.0) License used by PLOS journals. Please closely review the details of PLOSu2019s copyright requirements here: PLOS Licenses and Copyright. If you need to request permissions from a copyright holder, you may use PLOS's Copyright Content Permission form.

Potential Copyright Issues:

i) Please confirm (a) that you are the photographer of S1A-S10A, or (b) provide written permission from the photographer to publish the photo(s) under our CC BY 4.0 license.

3) Please amend your detailed Financial Disclosure statement. This is published with the article. It must therefore be completed in full sentences and contain the exact wording you wish to be published.

2) If any authors received a salary from any of your funders, please state which authors and which funders..

**Reviewers' Comments:**

Reviewer's Responses to Questions

**Key Review Criteria Required for Acceptance?**

**Methods**

-Are the objectives of the study clearly articulated with a clear testable hypothesis stated?

-Is the study design appropriate to address the stated objectives?

-Is the population clearly described and appropriate for the hypothesis being tested?

-Is the sample size sufficient to ensure adequate power to address the hypothesis being tested?

-Were correct statistical analysis used to support conclusions?

-Are there concerns about ethical or regulatory requirements being met?

Reviewer #1: Yes

Reviewer #2: -Are the objectives of the study clearly articulated with a clear testable hypothesis stated?

-Is the study design appropriate to address the stated objectives? Yes

-Is the population clearly described and appropriate for the hypothesis being tested? Yes

-Is the sample size sufficient to ensure adequate power to address the hypothesis being tested? Yes

-Were correct statistical analysis used to support conclusions? Yes but further explanation f qualitative methods needed

-Are there concerns about ethical or regulatory requirements being met? No concerns

Reviewer #3: The objectives are articulated with clear hypothesis and the study design is appropriate. A sample size calculation was not included. Although the size attained seemed reasonable. The statistical analysis, and conclusions are appropriate and there were no concerns regarding ethical standards

**Results**

-Does the analysis presented match the analysis plan?

-Are the results clearly and completely presented?

-Are the figures (Tables, Images) of sufficient quality for clarity?

Reviewer #1: Yes

Reviewer #2: -Does the analysis presented match the analysis plan? Yes

-Are the results clearly and completely presented? Yes but need direct quotes or more evidence for the qualitative findings

-Are the figures (Tables, Images) of sufficient quality for clarity? Yes

Reviewer #3: The analysis presented matches the plan and the results in the prose are clearly stated. The tables however are not very clear, specifllcally table 2. This has the question in the questionaire with the answer in bracket. It is difficult to follow. Perhaps sumarizing this as a figure with the actions instead being the subject of the question rather than the entire question.. to make this clearer to the reader. There are 8 tables which are cumbersome to read and could be converted to figures for ease of understanding

**Conclusions**

-Are the conclusions supported by the data presented?

-Are the limitations of analysis clearly described?

-Do the authors discuss how these data can be helpful to advance our understanding of the topic under study?

-Is public health relevance addressed?

Reviewer #1: Yes

Reviewer #2: -Are the conclusions supported by the data presented? Yes

-Are the limitations of analysis clearly described? Yes

-Do the authors discuss how these data can be helpful to advance our understanding of the topic under study? Yes

-Is public health relevance addressed? Yes

Reviewer #3: The conclusions are supported by the data and the limitations described. They do discuss how these data is helpful and public health relevance is adressed

**Editorial and Data Presentation Modifications?**

Reviewer #1: yes

Reviewer #2: (No Response)

Reviewer #3: I think a minor revision is needed

**Summary and General Comments**

Reviewer #1: 1. This study utilized an online questionnaire that respondents completed themselves. The data may be subject to social desirability bias, despite the authors' efforts to minimize it. Additionally, the study does not measure actual practices in the field.

2. The conclusions drawn are strong, but the suggestions for "structured training" and "curriculum integration" are quite typical. The Discussion section could explore more innovative, system-level, or technology-based solutions to the ongoing challenges noted, such as strong traditional beliefs and transportation issues.

3. The term "adequate knowledge" is defined as being in the 75th percentile of the composite score, which is a somewhat technical measure. Providing the raw score range or more context in the Results section could help non-technical readers understand this better.

4. It is suggested to refer from the paper: "Blockchain-Enabled Healthcare Optimization: Enhancing Security and Decision-Making Using the Mother Optimization Algorithm". This reference discusses blockchain in healthcare and improved decision-making. It supports the need for better awareness of national guidelines and emergency contact services by suggesting a secure and efficient data-sharing platform for public health management Suggested insertion: Conclusion/Future Work

5. The Discussion acknowledges barriers such as transportation challenges and traditional beliefs. It would be helpful to connect these barriers to technology-driven solutions suggested in other references, such as mobile education tools, visual aids, and secure data sharing. This would change the recommendation from general "training" to "structured training with technology and community involvement".

6. The Discussion section successfully explains the results and places them within the context of other countries facing similar challenges.

7. The point about cognitive dissonance, where knowledge clashes with harmful practices, is noteworthy and deserves further exploration. This could suggest the need for a behavioral or psychological aspect in the required training.

Reviewer #2: (No Response)

Reviewer #3: Overall a well done study on an important topic which is underreported, would commend the others for this study.

Comments

1. Although the sample size is large, perhaps having a justification of the sample size with a power calculation

2. Tables are difficult to follow. Suggest editing these and/or changing a few to figures to enhance readibility. Specifllcally tables 2 and 6. These have the questions used in the questionnaire with the answer in brackets. It is difficult to follow. Perhaps sumarizing this as a figure with the actions instead being the subject of the question rather than the entire question.. to make this clearer to the reader.

3. There are a few grammatical errors that need correction

4. For table 3: Chi square test is a measure of association, therefore the term "Deteminant" may not be an accurate term, rather associations in the bivariate analysis

PLOS authors have the option to publish the peer review history of their article (what does this mean? ). If published, this will include your full peer review and any attached files.

**Do you want your identity to be public for this peer review?** For information about this choice, including consent withdrawal, please see our Privacy Policy .

Reviewer #1: No

Reviewer #2: No

Reviewer #3: No

**Figure resubmission:**
---

## [Decision Letter · Decision Letter 1]

24 Feb 2026

Dear Dr. Dayasiri,

We are pleased to inform you that your manuscript 'A country-wide survey of knowledge and practice regarding prevention, snake identification and pre-hospital management of children with snakebites amongst Public Health Midwives in Sri Lanka' has been provisionally accepted for publication in PLOS Neglected Tropical Diseases.

Best regards,

Wuelton Monteiro, Ph.D.

Section Editor

Wuelton Monteiro

Section Editor

Shaden Kamhawi

co-Editor-in-Chief

Paul Brindley

co-Editor-in-Chief

Reviewer's Responses to Questions

**Key Review Criteria Required for Acceptance?**

**Methods**

-Are the objectives of the study clearly articulated with a clear testable hypothesis stated?

-Is the study design appropriate to address the stated objectives?

-Is the population clearly described and appropriate for the hypothesis being tested?

-Is the sample size sufficient to ensure adequate power to address the hypothesis being tested?

-Were correct statistical analysis used to support conclusions?

-Are there concerns about ethical or regulatory requirements being met?

Reviewer #1: I recommend accepting the manuscript. The authors have addressed all points raised by the reviewers. The revised manuscript is clearer and better organized than the original. Given these significant improvements and the relevance of the findings.

1. The authors responded to all reviewer concerns, especially clarifying the quantitative methods and removing any confusion between qualitative and quantitative data.

2. The results section is better organized, with complex tables changed into more understandable figures and the text simplified for easier reading.

3. The study's technical quality has improved with a detailed sample size and power calculation, confirming that the final sample of 1,706 participants provides enough statistical power for the analysis.

4. This study offers important new insights into neglected tropical diseases by presenting the first nationwide assessment of Public Health Midwives' knowledge gaps about pediatric snakebites in Sri Lanka.

5. The analysis uses clearer statistical terms, such as changing "determinants" to "associations," which accurately reflects the type of analysis performed.

6. The overall presentation has improved through careful editing, and the discussion now includes new, technology-based solutions for healthcare challenges at the community level.

Reviewer #3: This is okay

**Results**

-Does the analysis presented match the analysis plan?

-Are the results clearly and completely presented?

-Are the figures (Tables, Images) of sufficient quality for clarity?

Reviewer #1: Good

Reviewer #3: These are better

**Conclusions**

-Are the conclusions supported by the data presented?

-Are the limitations of analysis clearly described?

-Do the authors discuss how these data can be helpful to advance our understanding of the topic under study?

-Is public health relevance addressed?

Reviewer #1: Good

Reviewer #3: Conclusions have improved

**Editorial and Data Presentation Modifications?**

Reviewer #1: Good

Reviewer #3: (No Response)

**Summary and General Comments**

Reviewer #1: Good

Reviewer #3: (No Response)

PLOS authors have the option to publish the peer review history of their article (what does this mean? ). If published, this will include your full peer review and any attached files.

**Do you want your identity to be public for this peer review?** For information about this choice, including consent withdrawal, please see our Privacy Policy .

Reviewer #1: No

Reviewer #3: No

---

## [Editor Report · Acceptance letter]

Dear Dr. Dayasiri,

We are delighted to inform you that your manuscript, "A country-wide survey of knowledge and practice regarding prevention, snake identification and pre-hospital management of children with snakebites amongst Public Health Midwives in Sri Lanka," has been formally accepted for publication in PLOS Neglected Tropical Diseases.

Best regards,

Shaden Kamhawi

co-Editor-in-Chief

Paul Brindley

co-Editor-in-Chief
